# Epinephrine Infiltration of Adipose Tissue Impacts MCF7 Breast Cancer Cells and Total Lipid Content

**DOI:** 10.3390/ijms20225626

**Published:** 2019-11-11

**Authors:** Pierre Avril, Luciano Vidal, Sophie Barille-Nion, Louis-Romée Le Nail, Françoise Redini, Pierre Layrolle, Michelle Pinault, Stéphane Chevalier, Pierre Perrot, Valérie Trichet

**Affiliations:** 1INSERM, Université de Nantes, UMR1238, Phy-Os, Sarcomes osseux et remodelage des tissus calcifiés, F-44035 Nantes, France; pierre.avril.44@gmail.com (P.A.); luciano.vidal@univ-nantes.fr (L.V.); lrlenail@hotmail.com (L.-R.L.N.); francoise.redini@univ-nantes.fr (F.R.); pierre.layrolle@univ-nantes.fr (P.L.); valerie.trichet@univ-nantes.fr (V.T.); 2CRCINA, INSERM, Université d’Angers, Université de Nantes, F-44035 Nantes, France; sophie.barille@univ-nantes.fr; 3INSERM Université de Tours, UMR1069, Nutrition, Croissance et Cancer, F-37032 Tours, France; michelle.pinault@univ-tours.fr (M.P.); stephane.chevalier@univ-tours.fr (S.C.); 4CHU de Nantes, Service de Chirurgie Plastique et des Brûlés, F-44035 Nantes, France

**Keywords:** adipose tissue, breast cancer, epinephrine, breast reconstruction

## Abstract

Background: Considering the positive or negative potential effects of adipocytes, depending on their lipid composition, on breast tumor progression, it is important to evaluate whether adipose tissue (AT) harvesting procedures, including epinephrine infiltration, may influence breast cancer progression. Methods: Culture medium conditioned with epinephrine-infiltrated adipose tissue was tested on human Michigan Cancer Foundation-7 (MCF7) breast cancer cells, cultured in monolayer or in oncospheres. Lipid composition was evaluated depending on epinephrine-infiltration for five patients. Epinephrine-infiltrated adipose tissue (EI-AT) or corresponding conditioned medium (EI-CM) were injected into orthotopic breast carcinoma induced in athymic mouse. Results: EI-CM significantly increased the proliferation rate of MCF7 cells Moreover EI-CM induced an output of the quiescent state of MCF7 cells, but it could be either an activator or inhibitor of the epithelial mesenchymal transition as indicated by gene expression changes. EI-CM presented a significantly higher lipid total weight compared with the conditioned medium obtained from non-infiltrated-AT of paired-patients. In vivo, neither the EI-CM or EI-AT injection significantly promoted MCF7-induced tumor growth. Conclusions: Even though conditioned media are widely used to mimic the secretome of cells or tissues, they may produce different effects on tumor progression, which may explain some of the discrepancy observed between in vitro, preclinical and clinical data using AT samples.

## 1. Introduction

Adipose tissue (AT) is a biologically active tissue, which releases soluble growth factors (vascular endothelium growth factor, insulin growth factor) inducing tissue revascularization, but also produces hormones (leptin, adiponectin), cytokines (interleukin 6) and insoluble fatty acids, which all interact through complex networks within a tumor microenvironment. Different in vitro and pre-clinical studies have demonstrated that AT including mature adipocytes and stem cells, promotes the proliferation, invasion and survival of breast cancer cells through different secreted factors [1,2,3,4,5].

In contrast specific lipids content in peritumoral AT of breast cancer patients were correlated with therapeutic benefits. Decreased levels of two polyunsaturated n-3 fatty acids (n-3 PUFA), docosahexaenoic and eicosapentaenoice acids (DHA 22:6n-3 and EPA 20:5n-3), in peritumoral AT of women were associated with aggressive multifocal tumors compared to unifocal ones. Moreover it was shown that DHA and EPA decreased resistance of experimental mammary tumors to taxanes, anthracyclines or radio-therapy. These preclinical results are supported by the improved outcome of chemotherapy on metastatic breast cancer that was observed in a phase II clinical study including dietary supplementation with n-3 PUFA [6,7,8,9,10,11].

In addition, AT transfer does not increase the risk of recurrence of breast cancer as recently suggested by large retrospective clinical studies [12,13,14]. However one retrospective study observed that patients presenting either ductal or lobular intraepithelial neoplasia, had an increased risk of local events in the group who had undergone lipofilling [15]. The method to harvest AT is one of the most important steps governing the success of AT transplantation. Different methods have been described in the literature studying the variables (fat harvesting technique, infiltration solution, donor site, fat processing, etc.) that influence adipocyte survival and the AT engraftment [16,17,18,19]. However, the breast cancer recurrence risk related to the AT harvesting method has not yet been investigated. To harvest fat tissue, some surgical teams use for fat harvesting, an infiltration solution that contains epinephrine, to induce vasoconstriction [20], but it is worth noting that epinephrine may enhance lipolysis in AT [21].

In our study, the ephinephrine-lactated Ringer’s infiltrated solution adipose tissue conditioned medium (EI-CM) was tested on proliferative and quiescent human Michigan Cancer Foundation-7 (MCF7) breast cancer cells. The lipid composition of conditioned media of non-infiltrated or epinephrine infiltrated-AT from five donors was investigated. EI-CM and epinephrine-infiltrated adipose tissue (EI-AT) were injected into orthotopic induced breast carcinoma in athymic mouse.

## 2. Results

### 2.1. MCF7 Cell Proliferation was Enhanced by Epinephrine-Infiltrated Adipose Tissue Conditioned Medium

Proliferation of MCF7 breast carcinoma cells was analyzed in adherent culture conditions by measuring mitochondrial activity (WST-1 assay) and cell viability was controlled by cell counting with trypan blue staining. As cancer cells lost their adherence to plastic when whole epinephrine-infiltrated adipose tissue (EI-AT) was used, EI-AT was replaced by EI-CM to complement cell culture medium in order to mimic secreted factors by EI-AT. Before being treated, cells were cultured overnight (16 h) in a standard medium without fetal bovine serum (FBS) in order to observe a synchronized response to growth factor stimulation. FBS supplementation (10%) enhanced MCF7 mitochondrial activity up to 240% compared to that without FBS (Figure 1a). Similarly supplementation with 50% epinephrine-infiltrated adipose tissue conditioned medium (EI-CM) and 50% MEM α medium (0% FBS) from three different donors enhanced MCF7 mitochondrial activity from 150 to 200% (Figure 1a). The increases of mitochondrial activity were correlated with an increase in cell number by trypan bleu counting. Moreover, independent experiments performed with 50% or 25% of EI-CM of patients n°1 to 3 showed similar increases of MCF7 cell proliferation. The inhibition of the ERK1/2 signaling pathway using 5 µM UO126 induced a 30% decrease of the MCF7 proliferation with EI-CM, whereas it did not change the MCF7 proliferation with 10% FBS (Figure 1b). These results indicate that EI-CM increases MCF7 cell proliferation at least partially through the ERK1/2 signaling pathway.

Cell distribution in each cell-cycle phases was observed by flow cytometry after DNA staining with propidium iodide. During culture without FBS, at least half of the MCF7 cells were in G_0_/G_1_ phase (54% in Figure 1c, top panel). FBS treatment decreased the proportion of cells in G_0_/G_1_ phase by half and increased the proportion of cells preparing their mitosis and those replicating their DNA (Figure 1c, middle panel). When MCF7 cells were treated with 25% EI-CM (Figure 1c, low panel), the proportion of cells in G_0_/G_1_ phase was also reduced by half compared to 0% FBS culture condition. With 25% EI-CM, a higher increase in cells in G_2_/M phase was observed compared to 10% FBS (plus 20% versus plus 11%) whereas the increase of cell proportion in S phase was weaker than with 10% FBS (plus 4% versus plus 15%). These results indicate that EI-CM complementation induced cell-cycle activation in MCF7 cells allowing cells to reach the G_2_/M phase faster than FBS complementation.

### 2.2. MCF7 Cell Quiescence was Increased by Sphereoid Culture and Reduced by Epinephrine-Infiltrated Adipose Tissue Conditioned Medium

Cell culture under anchorage-independent conditions induces carcinoma cells to form spheres and to undergo epithelial mesenchymal transition (EMT) which may correlate with a more invading phenotype such as carcinoma stem cells [22,23]. From MCF7 spheres, messenger ribonucleic nucleic acids (mRNAs) were isolated for relative gene expression analysis after three days in culture. Five genes *MYC*, *CD44*, *TWIST1*, *TWIST2* and *SNAI1* (official symbols and full gene names presented in Table 1) which are activated in breast carcinoma stem cells and during EMT exhibited a higher expression in MCF7 cells cultured as spheroids (3-D) compared to that in MCF7 cells cultured in monolayer (2-D) (Figure 2a). In accordance with EMT, E-cadherin gene (*CDH1*) expression was lower in MCF7 spheroids than in monolayers. EI-CM treatment of 3-D cultured MCF7 cells increased expression of *MYC* and *TWIST2* while it decreased that of *CD44* and *SNAI1*, suggesting that the effect of EI-CM on EMT in MCF7 cells could be either as an activator or inhibitor of the EMT.

Tumor recurrence can be explained by the persistence of cancer cells in a quiescent/non-dividing stage that enables them to escape to chemotherapy agents during treatment, whereas microenvironment changes may later activate a cellular switch towards cell division. Quiescent cells corresponding to cells in G_0_ phase do not express the Ki-67 protein which is strictly associated with dividing cells in the G_1_, S, G_2_ or M phases [24]. Under anchorage-independent conditions (3-D), 26.9% of MCF7 cells were in G_0_ phase (Figure 2b, left panel). In 3-D culture conditions, 10% FBS complementation did not change cell distribution (Figure 2b, middle panel). In contrast, EI-CM complementation of 3-D cultured MCF7 cells induced a decrease of cell proportion in G_0_ phase (13.9%; Figure 2b, right panel).

Immunohistochemistry (IHC) targeting the Ki-67 protein was performed on spheroids formed by MCF7 cells cultured under anchorage-independent conditions (Figure 2c). Consistent with flow cytometry, 25% EI-CM treatment induced an increase of the Ki-67 positive cell proportion (+25%; Figure 2d left panel). Despite cell cycle activation by EI-CM, sphere number and volume (Figure 2d, right panel) were not significantly different between 1% FBS and 25% EI-CM complementation. Altogether, this indicates that EI-CM enabled G_0_ to G_1_ phase transition of MCF7 cells.

### 2.3. Epinephrine Infiltration Changed Lipid Content and Proliferative Effect of Adipose Tissue Conditioned Medium

For patients n°4 to 10, two AT samples were successively collected: a first one without epinephrine infiltration and a second one following infiltration with the epinephrine-lactated Ringer’s solution (ELR) and were used to obtain, respectively, NI-CM and EI-CM. As previously observed with EI-CM from patients n°1 to 3, the CM obtained from EI-AT samples had an increased proliferative effect on MCF7 cells, whereas their counterpart CM obtained with non-infiltrated AT did not (Figure 3a, left panel). As shown in Figure 3a (right panel), the ELR by itself had no effect on MCF7 cell proliferation. Because epinephrine infiltration may modify the metabolite composition of the AT samples through lipolysis enhanced by beta-adrenergic receptor activation, the comparison of lipid contents between NI-CM and EI-CM was performed to identify potential molecular mediators leading to EI-CM pro-proliferative effects. We compared the fatty acid content of AT-CM samples that were obtained from 5 donor sites either non-injected (NI) or ephinephrine-lactated Ringer’s infiltrated (EI). EI-CM showed a higher total lipid content compared to NI-CM of the corresponding donors (Figure 3b). This result suggested that EI-CM and NI-CM may present a different lipid content; however, the percentages of saturated, mono-unsaturated or polyunsaturated fatty acids (PUFAs n-3 and n-6) were similar (Figure 3c) and there were no statistically significant differences in individual fatty acid between EI-CM and NI-CM samples which were derived from 5 donor sites either infiltrated or non-infiltrated with ELR.

### 2.4. Injection of Epinephrine-Infiltrated Adipose Tissue or Corresponding Conditioned Medium into MCF7 Tumor in Mice

We were able to compare EI-AT and EI-CM injection in a preclinical model of breast carcinoma. Orthotopic breast carcinoma were induced in athymic mice by intraductal injection of MCF7 cells and a single injection of either PBS, EI-CM or EI-AT was performed at the tumor site after 90 days when tumors were detectable (>70 mm^3^).

PBS-treated group (Figure 4b, top panel) showed a slow tumor development, reaching a mean volume of 200 mm^3^ at day 160. Human Ki-67 protein detection confirmed the presence of tumor cells in mammary ducts (Figure 4a, top panel) as well as in surrounding adipose and connective structures (Figure 4a, middle panel). These observations may indicate that the tumor first grew within mammary ducts before invading the rest of mammary fat pad, as a ductal carcinoma in situ that would have turned invasive.

Two of six tumors in the EI-CM-injected group showed faster development compared to tumors of the PBS-injected group (Figure 4b, top and middle panels, respectively); however differences between the tumor volume means of these two groups were not significant at day 160. Tumor growth was more similar between PBS-and EI-AT-injected groups (Figure 4b, top and low panels, respectively) than between PBS- and EI-CM-injected groups. However the percentages of Ki-67 positive cells ranging from 18 to 26% were not significantly different between EI-AT-, EI-CM- and PBS-treated groups as determined after human Ki-67 protein immunohistochemical staining on tumor samples (Figure 4a). These results indicate that EI-CM may have slightly (but not significantly) promoted MCF7 tumor growth while corresponding whole EI-AT may not have modified breast tumor growth.

## 3. Discussion

AT transplantation has become an increasingly common technique in aesthetic breast augmentation, in non-oncological and in oncological breast reconstruction [25]. Low complication rates, readily available donor sites with low donor-site morbidity and an aesthetic benefit are some of the advantages of the AT transfer. Although AT transplantation has proven effective in breast reconstruction, safety concerns have been raised regarding its use in patients with a history of breast cancer [26,27,28]. At present, large cohort retrospective studies or systematic literature reviews and meta-analyses suggest that AT transplantation is safe with no increase of cancer recurrence risk for breast cancer patients after treatments [29,30,31,32,33]. However AT harvesting procedures have been poorly described in these clinical studies, whereas the lack of standardized protocols for harvesting may explain unpredictable clinical outcomes with AT engraftment [34,35,36].

Ephinephrine-lactated Ringer’s solution is often used to infiltrate AT before harvesting, in order to reduce bleeding, but it may modify the metabolite composition, especially cholesterol and fatty acids, of AT samples since catecholamines induce lipolysis in adipocytes [37,38,39]. Interestingly, polyunsaturated n-3 fatty acids in peritumoral AT of breast cancer patients may have beneficial effects on the disease progression.

In this study, we sought to understand how soluble factors secreted by EI-AT may influence the proliferation and quiescent state of breast cancer cells. EI-AT secreted factors that were collected in the conditioned culture medium induced a significant increase in the proliferation rate (150% to 200%) of MCF7 breast cancer cells, while non-infiltrated AT soluble factors did not. EI-AT secreted factors increased MCF7 cell proliferation at least partially through the extracellular-regulated kinase (ERK) 1/2 signaling pathway. In a previous study using similar EI-CM samples, multi cytokine assay has identified interleukin 6 (IL-6) and leptin as molecular candidates to induce increase of osteosarcoma cell proliferation; however neither IL-6 nor leptin have been able to mimic the pro-proliferative effects of EI-CM. By in vitro and preclinical studies, Danilo C. et al. have shown that cholesteryl ester via its cellular receptor (scavenger receptor class B type I, SR-BI) increase breast cancer cell proliferation through the phosphatidylinositol 3-kinase (PI3K)/protein kinase B (AKT) pathway but not through the mitogen-activated protein kinase (MAPK)/ERK1/2 pathway [40]. Despite the important role of ERK1/2 in the proliferation of breast cancer cells in vitro, activation of ERK1/2 was not associated with enhanced proliferation and invasion of 148 clinical mammary carcinomas [41].

During clinical procedures, EI-AT transplantation is never performed in a tumor site with proliferative cancer cells. Plastic surgery is performed following a cancer-free period and at worst, the primary tumor site may contain quiescent/dormant cancer cells. The quiescent state of breast cancer cells in vitro was induced by culture under anchorage-independent conditions, using methylcellulose in the culture medium. We observed that EI-CM induced an output of the quiescent state of MCF7 cells when maintained in non-adherent spheres: 14% of cells in G_0_ phase with EI-CM compared to 27% or 24% in 1% and 10% FBS supplementation, respectively. Such 3-D culture conditions induced a slight change from epithelial towards mesenchymal phenotypes of MCF7 cells, as suggested by *MYC, CD44, TWIST1/2* and *SNAI1* expression increase associated with a decrease of *CDH1* expression. We observed that EI-CM did not enhance such potential EMT in MCF7 cells. In this study, we did not test EI-CM effect on the migration of breast cancer cells and we did not use primary breast cancer stem cells which are of high interest in the progression, treatment resistance and recurrence. Originally, Charvet H.J. et al. have used one breast cancer cell line derived from one out of 10 patient specimens, and not a purchased banked cell line. Charvet H.J. et al. showed a 10-fold migration increase of primary breast cancer cells when cocultured with adipose-derived stem cells isolated from the same patient.

To conduct in vitro assays, an AT-conditioned medium (AT-CM) is usually used in the literature, instead of the whole AT sample itself which would be injected into a patient. AT-CM contains AT secreted and soluble factors including growth factors, cytokines and free fatty acids, but no adipocytes or adipose-derived stem cells. Dirat B. et al. have demonstrated that adipocytes obtained from breast AT during tumorectomy, exhibit a loss of lipid content, an expression increase of proinflammatory cytokines and the ability to increase invasive capacities of breast cancer cell lines.

Conditioned media derived from epinephrine-infiltrated AT showed a pro-proliferative effect on breast cancer cells and significantly higher lipid contents compared to non-infiltrated AT of corresponding patients. However, the percentages of saturated, mono-unsaturated or polyunsaturated fatty acids were similar in EI- or NI-CM. Recently, Wang Y.Y. et al. showed that free fatty acids released from adipocytes were incorporated into breast cancer cells as triglycerides in lipid droplets and that saturated fatty acids but not unsaturated ones were increased in cocultured cells [42]. Additionally, they demonstrated that lipolysis in adipocytes was induced by tumor cell secretions, but was not induced by catecholamines. In our study, only patients with a standard body mass index ranging from 20 to 22 were included. Because obesity is clearly related to a higher risk of cancer [43], including breast cancer risk, it would be of interest to compare the total lipid contents of EI-CM derived from obese and lean patients.

Epinephrine-infiltrated adipose tissue-conditioned medium (EI-CM) and whole epinephrine-infiltrated adipose tissue (EI-AT) of the same patient were compared following a single injection within breast carcinomas induced in athymic mice by intraductal injection of MCF7 cells. Interestingly, carcinoma growth was slow in this preclinical model; tumors were visible only 90 days after MCF7 cell injection, despite implantation of pellet delivering 17β-estradiol, and tumor volumes were less than 400 mm^3^ two weeks after EI-AT or EI-CM single injections. A single injection of EI-CM may have slightly but not significantly increased MCF7 tumor growth compared to a single PBS injection. The untranslation of the EI-CM cellular effects to in vivo effects may be due to reversed or transient effects that have not been tested in our in vitro study, or due to neutralization through molecular interactions with physiological liquids. In contrast, corresponding whole EI-AT injection did not modify breast tumor growth, in agreement with clinical studies which show that AT transfer did not increase tumor recurrence and then, may have no effect on quiescent tumor cells.

MCF-7 cell line which is a luminal A subtype of breast cancer expressing estrogen and progesterone receptors, is not the more aggressive and invasive model. It will be of high interest to test a model with a higher metastatic potential such as the MDA-MB-231 cell line, a basal subtype of triple negative breast cancer. We observed that EI-CM of patients n°1 to 3 induced 50% increase in the proliferation rate of MDA-MB-231 cells in culture (data not shown), but we did not investigate further this cell line as we were not able to establish an adequate in vivo model using it. A low tumor incidence (50%) with high variability of tumor size was obtained using MDA-MB-231 cell injection in nude mice.

In conclusion, epinephrine-infiltration of AT induces secretion of factors including lipids. This may contribute to the pro-proliferative effect and output of the quiescent state that were observed in vitro on MCF7 breast cancer cells. Moreover, such epinephrine-induced secreted factors seem to increase the in vivo growth of MCF7-induced tumor in mice. However, a single injection of whole epinephrine-infiltrated AT did not increase the slow progression of MCF7-induced tumor in mice, revealing a discrepancy between the effects of AT-secreted soluble factors in the conditioned medium and the whole AT sample which would be injected into a patient. The proportion of polyunsaturated fatty acids was not modified by the epinephrine infiltration, despite the significant increase in secreted lipids by EI-AT. The results of the EI-AT presented here do not call into question the safety of AT transplantation, however it would be of interest to compare cancer recurrences in breast cancer patients following the transplantation of AT harvested with or without epinephrine infiltration.

## 4. Materials and Methods

### 4.1. Adipose Tissue (AT)

AT samples: Human adipose tissue (AT) was obtained from abdominal liposuction during plastic surgery at the University Hospital of Nantes. Donors (patients n°1 to 10) with no significant medical history gave informed consent for the use of surplus AT sample for anonymized unlinked research, as validated by the “Comité de Protection des Personnes des Pays de la Loire” and by the “Ministère de la Recherche” (Art. L. 1245-2 of the French public health code, Law no. 2004-800 of 6 August 2004, Official Journal of 7 August 2004) with declaration to the “Commission Nationale de l’Informatique et des Libertés”. Ten AT samples were collected using a 12-gauge, 12-hole cannula (Khouri Harvester) connected to a 10 mL Luer-Lock syringe after infiltration with 0.1% epinephrine lactated Ringer’s solution as performed in our department to reduce bleeding. All patients had a body mass index ranging from 20 to 22. For five patients, we obtained both epinephrine infiltrated (EI) and non-infiltrated (NI) samples. Samples were centrifuged at 3000 rpm with a 9.5 cm radius fixed angle rotor for 1 min (Medilite™, Thermo Fisher Scientific, Illkirch, France) at room temperature. After centrifugation, the samples were separated into 3 layers: the upper one composed of oil, the middle one composed of the adipose tissue and the bottom one with blood and infiltration solution. Only middle layers corresponding to AT samples were collected.

AT-Conditioned Medium (AT-CM): AT samples were placed in cell culture inserts (pore size 3 µm; Becton Dickinson, Le Pont de Claix, France) with Minimum Essential Medium alpha (Gibco^®^ MEM α; Life technologies, St Aubin, France) with nucleosides and 1 g/L D-Glucose (MEM) under serum-free conditions. After 24 h, inserts with AT were removed and AT-CM was collected and frozen at minus 20 °C.

### 4.2. Culture Conditions

MCF7 cells were initially derived from a human breast adenocarcinoma ATCC number HTB-22, (ATCC, Manassas, VA, USA). They are a luminal subtype and express estrogen, progesterone and glucocorticoid receptors. MEM α medium was supplemented with 10% fetal bovine serum (FBS) and used to culture cells at 37 °C in a humidified atmosphere (5% CO_2_/95% air). For culture under anchorage-independent conditions, 1 mL MEM α medium was supplemented with 1.05% of methylcellulose (R&D Systems, Lille, France) and 1% FBS, and was seeded with 1 × 10^5^ cells into a well of 24-well plate for 3 days. Then 0.5 mL MEM α medium supplemented with 1% or 10% FBS or with 25% EI-CM were added for 2 days. Complete FBS starving (0%) was avoided in 3-D culture to maintain a low proportion (<5%) of cells in subG0 phase.

### 4.3. Cell Viability

Three thousand cells per well were cultured into 96-well plates with medium supplemented with FBS, CM or chemical inhibitor of ERK1/2 phosphorylation, UO126 (R&D Systems). After 24 h, WST-1 reagent (Roche Diagnostics, Meylan, France) was added to each well for 2 h at 37 °C. Then absorbance was read at 450 nm.

### 4.4. Cell Cycle Analysis

MCF7 cells were obtained and analyzed as previously described [44]. Briefly, DNA was stained in ethanol-fixed cells with propidium iodide (50 µg/mL; Sigma-Aldrich, Lyon, France) and Ki-67 protein was eventually detected using a FITC-coupled mouse anti-human Ki-67 antibody (Becton-Dickinson, Le pont de Claix, France). Cell fluorescence was measured by flow cytometry (Cytomics FC500; Beckman Coulter, Villepinte, France). Cell-cycle distribution was analyzed for 20,000 events using MultiCycle AV Software, Windows version (Phoenix Flow System, San Diego, CA, USA) to obtain histograms of cell repartition in each cell-cycle phase and CXP Analysis software version 2.2 (Beckman Coulter, Villepinte, France) to obtain dot-plots of DNA/Ki-67 double-staining.

### 4.5. Reverse Transcription and Quantitative PCR

Gene expression was observed as previously described [45], after RNA extraction (NucleoSpin RNA II; Machery-Nagel, Düren, Germany), reverse transcription with ThermoScript RT (Invitrogen Life Technologies, Villebon sur Yvette, France) and cDNA amplification using the IQ SYBR Green Supermix (Bio-Rad, Marne la Coquette, France). For quantitative analysis, the iCycler iQ Real-time PCR Detection system (Bio-Rad), was used to calculate relative fold change of gene expression, following the delta delta Ct method [46]. The *HPRT1* reference gene was used for normalization. Primer sequences with corresponding gene symbol and name are indicated in Table 1.

### 4.6. Breast Carcinoma Model

Eight-week-old female athymic mice (NMRI nu/nu) were obtained from Elevages Janvier (Le Genest St Isle, France). They were housed under pathogen-free conditions at the Experimental Therapy Unit (Faculty of Medicine, Nantes, France). The experimental protocol was approved by the regional committee on animal ethics (CEEA.PdL.06) and the Minister of Agriculture (Authorisation number: 9634) and was conducted following the guidelines “Charte nationale portant sur l’éthique de l’expérimentation animale” of the French ethical committee. The mice were anaesthetized by inhalation of an isoflurane-air mix (2% for induction and 0.5% for maintenance, 1 L/min) before injection with 2 × 10^6^ MCF7 cells in 30 µL of Matrigel (R&D Systems) diluted in phosphate buffered saline (PBS 50%) into the 4th left mammary duct. At the time of cell injection, a pellet delivering 17β-estradiol (Innovative Research of America, Sarasota, FL, USA) was subcutaneously implanted between the neck and the left shoulder. Formula (l^2^xL)/2, where l and L represent the smallest and largest diameter respectively, was used to calculate the tumor volume [47].

### 4.7. Histology Analysis

Tumor samples were fixed in 4% buffered paraformaldehyde (PFA) for 48 h, while sphere samples were fixed in 4% PFA for 15. Three µm-thick sections of tumors or spheres embedded in paraffin were dewaxed, rehydrated and then treated with 3% H_2_O_2_ for 15 min at room temperature. Human Ki-67 immunohistochemistry detection was then performed with a mouse monoclonal anti-human Ki-67 (MIB-1 clone; Dako, Les Ulis, France) and revealed with a biotinylated goat anti-mouse Immunoglobulin G secondary antibody and Streptavidin-Horse Radish Peroxydase complexes (Dako) that were observed following an incubation with 3,3′-Diaminobenzidine (DAB, Dako). Nuclei were counterstained with a Gill-Haematoxylin solution. ImageJ software (NIH, Bethesda, MD, USA) was used to calculate the proportion of Ki-67-positive cells, from counting >15,000 nuclei in 6 sections of each tumor sample or >5000 nuclei in 6 sections of each sphere sample.

### 4.8. Lipid Analysis

Fatty acid composition analysis using gas chromatography: AT-CM were frozen in liquid nitrogen before total lipid extraction according to the FOLCH method with chloroform-methanol 2:1 (*v*/*v*). Extracts from AT-CM were washed with saline and separated into two phases. The chloroform phase transferred to a new tube was evaporated. Lipid extracts were resuspended with 200 µl of chloroform-methanol 2:1 (*v*/*v*). Triglycerids (TG) were separated on silica gel TLC plates (LK5, 20*20; Merck St Quentin, Yvelines, France) for thin layer chromatography. After spot scraping, TG were collected and treated as fatty acids methyl esters (FAME) for gas chromatography analysis.

Derivatization of fatty acids was performed with 14% boron trifluoride (in methanol), which resulted in the formation of methyl esters. The derivatization mixture was incubated and shaken for 30 min at 100 °C. Finally, FAME were extracted twice with hexane and then evaporated to dryness. Batch samples were analzsed with a gas chromatograph (GC-2010plus; Shimadzu Scientific instruments, Noisiel, France) through a capillary column (SGE BPX70 GC Capillary Columns; Chromoptic SAS, Courtaboeuf, France). A hydrogen carrier gas was maintained at 120 kPa. Oven temperature was set to 60 °C to 220 °C and flame ionization detector temperature at 280 °C for fatty acid detection. FAME identification was done by comparing relative retention times of samples to those obtained for pure standard mixtures (Supelco 37 fatty acid methyl Ester mix; Sigma Aldrich). The relative amount of each fatty acid (saturated, mono-unsaturated or polyunsaturated fatty acid) was quantified by integrating the baseline peak divided by the peak area corresponding to all fatty acids, using the GC solutions software (Shimadzu Scientific instruments, Noisiel, France).

### 4.9. Statistical Analysis

Microsoft Excel software (Redmond, WA, USA) was used. In vitro experiments results were analyzed following the analysis of variance t-test. In vivo experimentation results were analyzed with the unpaired nonparametric method and Dunn’s multiple comparisons following the Kruskal-Wallis test.

## Figures and Tables

**Figure 1 ijms-20-05626-f001:**
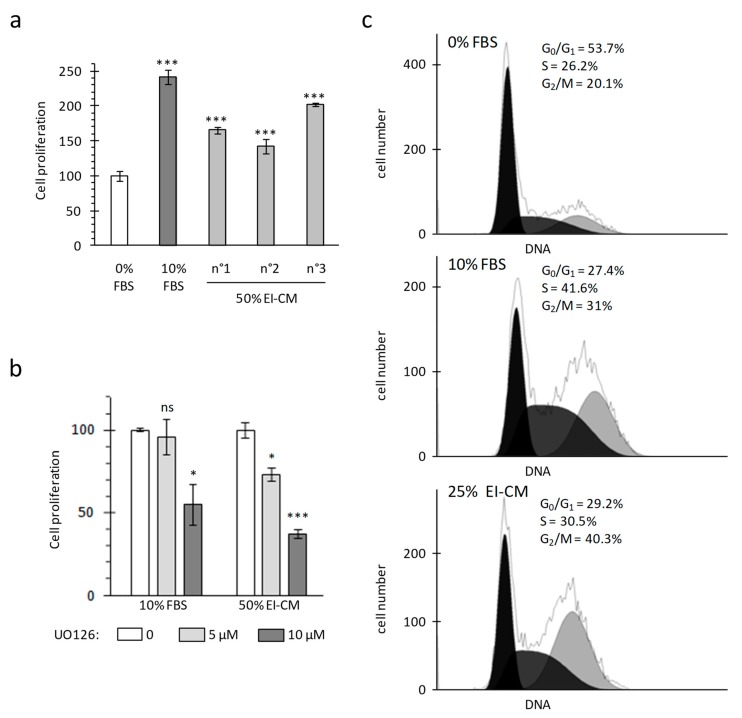
Michigan Cancer Foundation-7 (MCF7) cell growth with ELR solution-infiltrated adipose tissue conditioned medium (EI-CM). MCF7 cells were cultured during 24 h in a medium without any growth factor (0% FBS) or supplemented with fetal bovine serum (10% FBS) or EI-CM (25 to 50%) and MEM with 0%FBS. (**a**) Histogram shows the mitochondrial activity of MCF7 cells, measured by WST-1 assay. EI-CM were derived from 3 different donors (n°1 to n°3). Results are the means of 3 wells and are presented as a percentage of 0% FBS value with standard deviations. Statistically significant differences are indicated in comparison with 0% FBS (***: *p* < 0.0001)**.** Each patient EI-CM was tested in 3 independent experiments. (**b**) Mitochondrial activity of MCF7 cells measured by WST-1 assay. Cells were cultured for 24 h with or without 5 or 10 µM of ERK inhibitor UO126. Results are the means of 3 wells and are presented as a percentage of condition without UO126 with standard deviations. Statistically significant differences are indicated in comparison with 0 UO126 (*: *p* < 0.05; ***: *p* < 0.0001). Two independent experiments were performed. (**c**) Histograms show the distribution of MCF7 cells in cell-cycle phases following DNA detection by flow cytometry. Because only 2–3% of cells were identified in the subG_0_ phase, only the proportion of cells in the G_0_/G_1_, S and G_2_/M phases are indicated.

**Figure 2 ijms-20-05626-f002:**
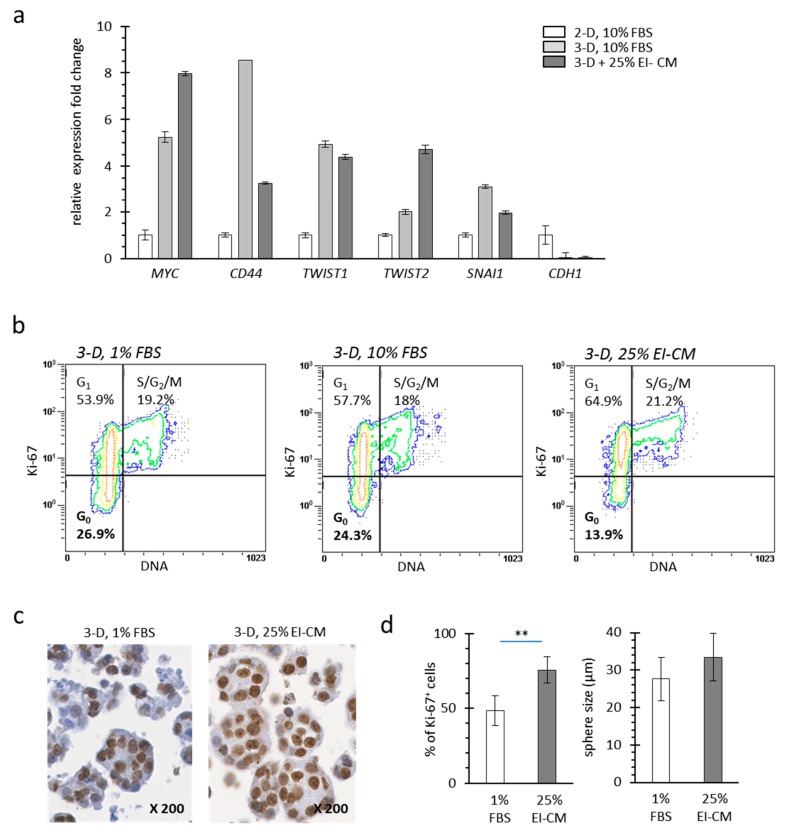
Quiescence of MCF7 cells with ELR solution-infiltrated adipose tissue conditioned medium (EI-CM). (**a**) Relative expression fold changes are presented for mRNA of MCF7 cells cultured either in monolayer (2-D) or in sphere (non-anchorage conditions, 3-D) during 3 days and treated 48 h in a medium supplemented with 1 or 10% FBS or with 25% EI-CM. Expression change of those 5 genes has been correlated to epithelial mesenchymal transition leading to invading phenotype of carcinoma cells. Full gene names and symbols are indicated in Table 1. Means of 3 samples are presented with standard deviations. Significant differences are not indicated as this experiment was not repeated. (**b**) Dot plots show the distribution in cell cycle phases (G_0_, G_1_ and S/G_2_/M) following DNA and Ki-67 detection in MCF7 cells cultured in spheres (3-D). Because only 2–3% of cells were identified in subG_0_ phase, only the proportion of cells in G_0_, G_1_, and S/G_2_/M phases are indicated. (**c**) Representative images of IHC detection of Ki-67 on MCF7 spheres (3-D). Magnification is indicated. (**d**) Histograms show the Ki-67 index (left panel) and the mean diameter (right panel) of MCF7 spheres. In the left panel, percentages were counted on 6 representative regions for each treatment after Ki-67 detection by IHC. In the right panel, mean diameter was calculated on 90 different spheres for each condition. Statistically significant differences are indicated in comparison with 1% FBS (**: *p* < 0.001). Three independent experiments were performed.

**Figure 3 ijms-20-05626-f003:**
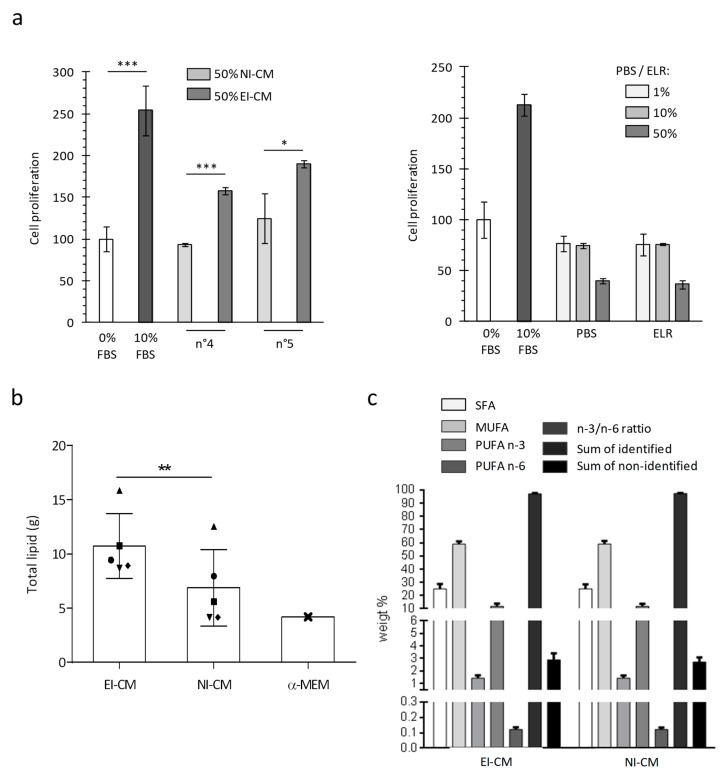
MCF7 cell growth and lipid composition depending on ELR solution infiltration of harvested adipose tissue. (**a**) Histogram shows the mitochondrial activity of MCF7 cells measured after 24 h. For the left panel, cells were cultured in medium without FBS (0% FBS) or supplemented with 10% FBS or with 50% EI-CM and 50% MEM α with 0%FBS. AT samples were obtained from 2 different donors (n°4 and n°5) and were initially not-infiltrated (NI) or infiltrated with ELR (EI). For right panel, cells were cultured in a medium without FBS (0% FBS) or supplemented with 10% FBS, PBS or ELR, representing 1 to 50% of the total volume. *: *p* < 0.05; ***: *p* < 0.0001. (**b**) Histogram shows the total lipid amount detected in conditioned medium from infiltrated with ELR (EI-CM) or not-infiltrated AT (NI-CM) for 5 patients (n°6 to n°10) who are represented by a distinct geometric forms. Lipid amount is indicated in standard culture medium without FBS (MEM α). **: *p* = 0.0045 paired t-test. (**c**) Histogram shows the weight % of fatty acids derived from 5 donors either infiltrated or non-infiltrated with ELR. Saturated, mono-unsaturated or polyunsaturated fatty acids (SFA, MUFA or PUFA) were measured in a conditioned medium of epinephrine lactated Ringer’s solution-infiltrated or non-infiltrated adipose tissue (EI-CM or NI-CM).

**Figure 4 ijms-20-05626-f004:**
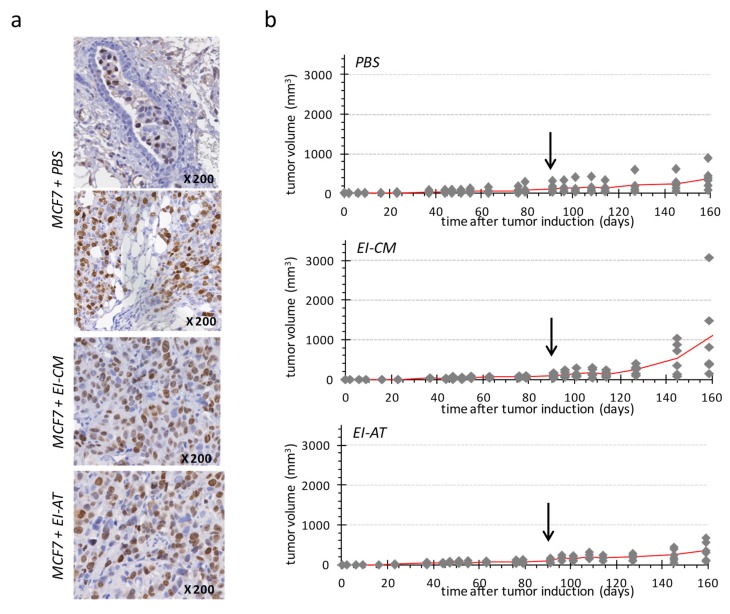
Single injection of either EI-AT or EI-CM in MCF7 tumor induced in athymic mouse. (**a**) Immunohistochemical staining of human Ki-67 protein in MCF7 tumors injected with PBS, EI-CM or EI-AT. Observation of mammary duct (top panel) and mammary fat pad (middle panel). (**b**) Evolution of tumor volume is reported for each group of mice (*n* = 6); the mean tumor volume is represented by a red line. The arrows indicate time of the intra-tumor injection of either PBS (top panel), or EI-CM (middle panel group) or EI-AT (low panel). At days 145 and 160, no significant difference between groups was detected by unpaired nonparametric method.

**Table 1 ijms-20-05626-t001:** List of genes analyzed by real time RT-PCR: Genes are presented with official gene symbols and corresponding full name. Forward and reverse primer sequences used to perform the analyses are indicated.

Official Symbol	Official Full Name	Reverse Primer
*HPTR1*	Hypoxanthine PhosphoRibosyl Transferase 1	CGAGCAAGACGTTCAGTCCT
*CD44*	Cluster of Differentiation 44	CGGCAGGTTATATTCAAATCG
*TWIST1*	Twist-related protein 1	TGCAGAGGTGTGAGGATGGTGC
*TWIST2*	Twist-related protein 2	AGAAGGTCTGGCAATGGCAGCA
*SNAI1*	Snail family transcriptional repressor 1	CAGCAGGTGGGCCTGGTCGTA
*CDH1*	Cadherin 1	CCAGCGGCCCCTTCACAGTC
*MYC*	Myelocytomatosis viral oncogene homolog	GATCCAGACTCTGACCTTTTGC

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
