# Peer review of "Epinephrine Infiltration of Adipose Tissue Impacts MCF7 Breast Cancer Cells and Total Lipid Content"

_ijms, 2019, doi:10.3390/ijms20225626_

Round 1

Reviewer 1 Report

REVIEW

The article is interesting and well written. It is necessary to improve the Discussion section reporting the different clinical application of fat tissue in the treatment of breast soft tissue defects. For this reason it is necessary to report, the results published in breast reconstruction, evaluating the safety of the fat graft into the breast through the Instrumental and histological evaluation. Also it is necessary to report the different methods to obtain fat graft enhanced with stromal vascular fraction cells (SVFs) and/or Platelet rich Plasma (PRP) and also the different fields in which the fat graft and SVFs can be used like wound healing when used alone or in combining with hyaluronic acid or for hair re-growth.

On this regard, the authors may add the following references:

(About the different methods to obtain SVFs in breast)

BreastReconstruction with Enhanced Stromal Vascular Fraction Fat Grafting: What Is the Best Method?

Gentile P, Scioli MG, Orlandi A, Cervelli V.

Plast Reconstr Surg Glob Open. 2015 Jul 8;3(6):e406. doi: 10.1097/GOX.0000000000000285. eCollection

(About hair regrowth) 

Autologous Cellular Method Using Micrografts of Human Adipose Tissue Derived Follicle Stem Cellsin Androgenic Alopecia.

Gentile P. Int J Mol Sci. 2019 Jul 13;20(14). pii: E3446. doi: 10.3390/ijms20143446.

Author Response

We are pleased to submit the manuscript « Epinephrine infiltration of adipose tissue impacts MCF7 breast cancer cells and total lipid content » revised according comments of editor and reviewers.

We would like to thanks the reviewers for their important work. All questions were of interest and useful to increase the quality of our results report.

As recommended by reviewer 1, we included 4 new references (14, 17, 18 and 19) describing clinical use and benefit of fat graft combined with stromal vascular fraction cells and/or platelet-rich plasma.

14. Gentile P, Casella D, Palma E, Calabrese C. Engineered fat graft enhanced with adipose-derived stromal vascular fraction cells for regenerative medicine: clinical, histological and instrumental evaluation in breast reconstruction. J Clin Med. 2019, 8, 504.

17. Gentile P, De Angelis B, Di Pietro V, Amorosi V, Scioli MG, Orlandi A, Cervelli V. Gentle Is Better: The original "Gentle Technique" for fat placement in breast lipofilling. J Cutan Aesthet Surg. 2018, 11, 120-126.

18. Gentile P, Orlandi A, Scioli MG, Di Pasquali C, Bocchini I, Curcio CB, Floris M, Fiaschetti V, Floris R, Cervell V. A comparative translational study: the combined use of enhanced stromal vascular fraction and platelet-rich plasma improves fat grafting maintenance in breast reconstruction. Stem Cells Transl Med. 2012, 1, 341-351.

19. Gentile P, Scioli MG, Orlandi A, Cervelli V. Breast reconstruction with enhanced stromal vascular fraction fat grafting: what is the best method? Plast Reconstr Surg Glob Open. 2015, 3, e406.

Reviewer 2 Report

     The current study aims to investigate the effect of the ephinephrine-lactated Ringer’s infiltrated solution adipose tissue conditioned medium (EI-CM) on proliferative and quiescent human MCF7 breast cancer cells. For their experiments, authors used  MCF7 cells cultured in monolayer and in 3-D conditions  and combined  to assess the objetive of this work. However, it would be highly appreciated if authors could clarify some points:

In the present study authors used MCF7 cells, a luminal A subtype breast cancer cell line, which is well characterized and broadly used. However, it has been described that MCF-7 is a poorly-aggressive and non-invasive cell line, normally being considered to have low metastatic potential compared with other breast cancer cell lines. Why have author chosen this type of breast cancer cell line? The results of this study claim that the culture of MCF-7cells in a EI-CM medium increases cell proliferation at least via ERK1/2 phosphorylation and cell cycle activation, in both monolayer and 3-D culture conditions. Have the authors checked if these effects are reversed when the cells are grown in a normal MEM medium again? How do the authors explain that EI-CM injection has not significantly effect tumor growth in MCF7 athymic mice compared to PBS injection?. In the section 2.4 authors indicate that the percentages of Ki-67 positive cells were not significantly different between the EI-AT and EI-CM treated groups. How different it is compared to PBS treated group? In addition, the fig 4a only shows Ki-67 detection in PBS treated mice. Images of EI-AT and EI-CM should be shown. In the western blot figures (figure 1b), the molecular weight of each band is not indicated. In addition, as it is mentioned in the instruction for the authors, the western blots images should be large enough to see the relevant features.   In the proliferation assay, or ERK phosphorylation assays (Fig 1a and fig 1b) cell are grown in a 50% EI-CM, while they are grown in a 25% EI-CM in cell cycle analysis (fig 1d), gene expression assay (fig 2a) or Ki-67 assay (fig 2d). What is the reason for this difference? Why is the ki-67 assay perfomed growing the cells only in a 1% FBS medium? In all other methods cell are grown in a 10% FBS medium. Authors should give a brief explanation why they have focused in the study of genes mentioned in fig 2a In the section 2.1, authors indicate that cell proliferation is measured by WST1-assay and cell counting with trypan blue staining. However this last method is not a cell proliferation assay. In addition, this methods are referred to as cell viability assay in the material and methods section (4.3). So, a specific section for cell proliferation assay is necessary Author contibutions section is not completed and well redacted.

Reviewer 3 Report

The manuscript of Vidal and collaborators evaluates the impact of adipose tissue harvested with or without epinephrine on the breast cancer cell line MCF7.

The authors prepared culture medium conditioned with human adipose tissue harvested in presence-or not, of epinephrine from 5 patients. They tested its effect on MCF7 cells proliferation and they also analyzed the lipid content of these media.

However, most experiments were performed by comparing the effects of conditioned medium from epinephrine-harvested adipose tissue to fetal bovine serum and not to conditioned medium obtained with adipose tissue harvested without epinephrine. Then the exact role of epinephrine is difficult to validate. Another major point of this paper is the absence of clear indication on the number of independent experiments that have been performed and that are used for statistical analysis. This raises a serious doubt on the relevance of the findings. At last, the benefits or risks of using epinephrine to harvest adipose tissue are not clearly stated and this reduces the impact of the conclusion. For instance, it may be very important to know if epinephrine will or will not promote significant tumor expansion when the harvested adipose tissue is used for lipofilling.

The following points have to be considered.

Figure 1 panels a & c. How many independent determinations were performed? Measurements from three wells within one experiment do not consist in independent determinations and cannot be used for valid statistical tests.

Figure 1b: Could the authors show the 10% FBS-induced pERK1/2 phosphorylation at 10 min as the weak signal at 30 min may just result from desensitization. How many times was this experiment repeated? In link with the results presented in figure 3, it would have be interesting to link this effect to a higher content of lipids or may be to any another molecule secreted in the conditioned medium.

Figure 1d: No comparisons with the NI-CM are shown. As the authors indicated that adipose tissue was obtained with or without epinephrine infiltration from the 5 patients the comparisons between the effects of NI-CM and EI-CM are required to properly evaluate the effect of epinephrine treatment.

Figure 1 & figure 3: The legend “% of the 0% FBS value” should be changed to be more easy to apprehend.

Figure 2a: Here again a comparison with the NI-CM has not been performed and is requested.

Figure 2b: Are the percentages of proliferating cells, i.e. cells in S/G2/M phases, significantly different in the three conditions tested? As they are in the same range of order, this may reflect a slower proliferation rate whatever culture condition used. This is in agreement with a volume of the spheres that does not differ significantly (Fig 2d). This is not consistent at all, with a higher Ki67 labelling (Fig 2d), which labels mitotic cells. How can the authors explain these results?

Figures. 2 c & d: Six representative regions analyzed form one experiment are not enough to present valid statistical analysis. How many independent experiments have been performed? I do not understand what the “1% FBS condition” represents exactly. Is it after 3 days of culture in methylcellulose? (see materials & methods page 10 lanes 31-35). Why the authors did not compare the cultures for 2 additional days with 10 % FBS and 25% EI-CM ?

Figure 3b: The amount of total lipid is quite different in the five patients. Can this amount be correlated to the BMI of the patients?

Figure 3c: From the text of the legend one may be confused as it seems that adipose tissue has been collected from 5 sites of the same donor and not collected from the same location in 5 patients as it is indicated in the legend of fig 3b and in the Materials and methods section. Could the authors correct this point?

Figure 4: Ki-67 expression reflects cells that are dividing, but it does not reflect any tumor potential. This labeling is not sufficient to ascertain that all cells dividing are only tumor cells. Double labelling with a specific protein expressed in MCF7 has to be performed.

Were the tumor volumes after 160 days significantly different depending on EI-CM or EI-AT injections?

Page 7, Lane 12, What Fig3b refers to?

Round 2

Reviewer 2 Report

Dear authors, I agree with your answers and with the new version of the manuscript.

Author Response

Dear reviewer number 2,

We would like to thanks you for your important work. All questions were of interest and useful to increase the quality of our results report.

Reviewer 3 Report

The revised version of the paper entitled « Epinephrine infiltration of adipose tissue impacts MCF7 breast cancer cells and total lipid content » is largely improved.

The authors gave explanations to most of the points raised. However few points still deserve attention, especially the point concerning Figure 1b.

INITIAL COMMENT : Figure 1b: Could the authors show the 10% FBS-induced pERK1/2 phosphorylation at 10 min as the weak signal at 30 min may just result from desensitization. How many times was this experiment repeated? 

YOUR ANSWER: Reviewer 3 was right by suggesting that ERK1/2 phosphorylation should be observed at earlier time in presence of EI-CM. In one of our unshown experiment (next figure), ERK1/2 phosphorylation was observed at 10 minutes with 10% FBS and became weak at 30 minutes (similarly to result shown in figure 1b). In this experiment, ERK1/2 phosphorylation was activated at 10 minutes with 50% EI-CM and maintained at 60 minutes and 120 minutes. Why this effect is maintained so long? Why it was not observed at 20 and 30 minutes in this experiment? In the manuscript, we presented only the part of the results that were obtained in independent experiments. In the 2.1 section, we added a comment on this important point that could be related to one Reviewer 2’s proposition: why not to test EI-CM withdrawal and see whether effect is reversed?

My answer to your comment: The blot that the authors presented in the answers to the Editor and reviewers, is far more convincing in terms of ERK 1/2 phosphorylation to me, as a 10 min stimulation with serum induced a very nice phosphorylation of ERK1/2, as expected. It would have been appropriate to show this figure with the corresponding loading controls instead of the one presented. Does it represent one representative experiment or has this experiment been repeated several times?

The decrease in the serum-induced ERK1/2 phosphorylation after 30 min of stimulation may result from ERK phosphatases activation. This may be one explanation that was previously reported (for instance, see Cook et al. 1997, J. Biol. Chem. Vol 272 pages 13309-19). This would just mean that the conditioned medium is less efficient to induce phosphatases expression than fetal bovine serum. If this is the case, it may present some interest.

INITIAL COMMENT :Figure 2a: Here again a comparison with the NI-CM has not been performed and is requested.

YOUR ANSWER: Experiments in figure 2 were only performed with NI-CM.

My answer to your comment: Sorry but the title of figure 2 is “Quiescence of MCF7 cells with ELR solution-infiltrated adipose tissue conditioned medium 2 (EI-CM)”. It was never indicated that NI-CM was used in this figure. In addition, your point by point answer to my queries mention that EI-CM was used in figures 1 & 2 (first answer). Could you please clarify?

INITIAL COMMENT : Figure 2b: Are the percentages of proliferating cells, i.e. cells in S/G2/M phases, significantly different in the three conditions tested? As they are in the same range of order, this may reflect a slower proliferation rate whatever culture condition used. This is in agreement with a volume of the spheres that does not differ significantly (Fig 2d). This is not consistent at all, with a higher Ki67 labelling (Fig 2d), which labels mitotic cells. How can the authors explain these results?

YOUR ANSWER: Both figures 2b and 2d present an increase of Ki-67 positive cells with EI-CM: flow cytometry detection showed +13% of Ki-67 positive cells, whereas immunohistochemical detection showed +25% of Ki-67 positive cells. Such differences may be explained by distinct process to access Ki-67 detection.

My answer to your comment: My point was much more to underline the discrepancy between KI-67 positive cells i.e. cells that will divide while the volume of the spheres is not changed. If there is a net proliferation, spheres should become larger. Is there any explanation for that?

INITIAL COMMENT: Figure 3b: The amount of total lipid is quite different in the five patients. Can this amount be correlated to the BMI of the patients?

YOUR ANSWER: We did not correlate total lipid in EI- or NI-CM to BMI of patients. In section material, we indicated that all patients have a BMI ranging from 20 to 22.  

My answer to your comment: This is very interesting as obesity is now recognized as a risk factor for cancer. Thus even from lean patients, conditioned medium has an impact on breast cancer cells. In this regard, the comparison of the effects of EI-CM from lean and obese patients would be of major interest.

INITIAL COMMENT: Figure 4: Ki-67 expression reflects cells that are dividing, but it does not reflect any tumor potential. This labeling is not sufficient to ascertain that all cells dividing are only tumor cells. Double labelling with a specific protein expressed in MCF7 has to be performed.

YOUR ANSWER: We indicated when information was missing, that the human Ki-67 protein is detected by immunohistochemistry. Mouse Ki-67 protein is not detected with the mouse monoclonal anti-human Ki-67 antobody that was used. The reference of the antibody was added in the materials and methods section. 

 My answer to your comment: this answer is fine even if I was thinking to another marker to specifically label MCF7 cells.
